# A Novel Sesterterpenoid, Petrosaspongin and γ-Lactone Sesterterpenoids with Leishmanicidal Activity from Okinawan Marine Invertebrates

**DOI:** 10.3390/md23010016

**Published:** 2024-12-30

**Authors:** Takahiro Jomori, Nanami Higa, Shogo Hokama, Trianda Ayuning Tyas, Natsuki Matsuura, Yudai Ueda, Ryo Kimura, Sei Arizono, Nicole Joy de Voogd, Yasuhiro Hayashi, Mina Yasumoto-Hirose, Junichi Tanaka, Kanami Mori-Yasumoto

**Affiliations:** 1Department of Chemistry, Biology and Marine Science, Faculty of Science, University of the Ryukyus, Nishihara, Okinawa 903-0213, Japanjtanaka@sci.u-ryukyu.ac.jp (J.T.); 2Faculty of Pharmaceutical Sciences, Tokyo University of Science, Noda, Chiba 278-8510, Japan; 3Faculty of Agriculture, University of Miyazaki, 1-1 Gakuen-kibanadai-nishi, Miyazaki 889-2192, Japanhayashi_yasuhiro@cc.miyazaki-u.ac.jp (Y.H.); 4Naturalis Biodiversity Center, 2300 RA Leiden, The Netherlands; nicole.devoogd@naturalis.nl; 5Institute of Environmental Sciences (CML), Leiden University, P.O. Box 9518, 2300 RA Leiden, The Netherlands; 6Tropical Technology Plus, Uruma, Okinawa 904-2234, Japan; myhiro@ttc.co.jp

**Keywords:** *Leishmania major*, furanosesterterpenoid, tetronic acid, γ-butenolide, marine sponge

## Abstract

Leishmaniasis is a major public health problem, especially affecting vulnerable populations in tropical and subtropical regions. The disease is endemic in 90 countries, and with millions of people at risk, it is seen as one of the ten most neglected tropical diseases. Current treatments face challenges such as high toxicity, side effects, cost, and growing drug resistance. There is an urgent need for safer, affordable treatments, especially for cutaneous leishmaniasis (CL), the most common form. Marine invertebrates have long been resources for discovering bioactive compounds such as sesterterpenoids. Using bioassay-guided fractionations against cutaneous-type leishmaniasis promastigotes, we identified a novel furanosesterterpenoid, petrosaspongin from Okinawan marine sponges and a nudibranch, along with eight known sesterterpenoids, hippospongins and manoalides. The elucidated structure of petrosaspongin features a β-substituted furane ring, a tetronic acid, and a conjugated triene. The sesterterpenoids with a γ-butenolide group exhibited leishmanicidal activity against *Leishmania major* promastigotes, with IC_50_ values ranging from 0.69 to 53 μM. The structure–activity relationship and molecular docking simulation suggest that γ-lactone is a key functional group for leishmanicidal activity. These findings contribute to the ongoing search for more effective treatments against CL.

## 1. Introduction

Leishmaniasis is a major public health problem affecting the world’s most vulnerable and poorest populations. It is caused by parasitic protozoa and is endemic in 90 tropical and subtropical countries. Worldwide, leishmaniasis is considered one of the top ten neglected tropical diseases, with 0.9 to 1.6 million new cases and 20,000 to 30,000 deaths each year [1]. Approximately 12 million people are currently infected and another 350 million are at risk. There are three forms of the disease: cutaneous, mucosal and visceral. Cutaneous leishmaniasis (CL) is the most common form, causing skin sores and ulcers that can lead to severe scarring and social stigma.

Current treatments, such as antimonial compounds and liposomal amphotericin B, have problems including high toxicity, serious side effects, high cost, and increasing drug resistance [2,3]. The oldest chemotherapies used for leishmaniasis are antimonial compounds, which are administered intravenously or intramuscularly. These compounds contain heavy metals that cause side effects such as nausea, vomiting, headache, and, occasionally, cardiotoxicity. In addition, some parasites have developed resistance to these drugs [4]. Liposomal amphotericin B, a currently widely used intravenous formulation, reduces nephrotoxicity through liposome encapsulation and is designed to remain at the site of the lesion for prolonged periods. Miltefosine, an oral drug also used for amoebic infections, is used for all three forms of leishmaniasis, but its efficacy is variable and it is contraindicated in pregnant women [4]. These drugs are sometimes used in combination. However, these treatments are all expensive and not easily accessible to patients in endemic regions, leading most people to abandon treatment. Treatment of CL is also challenging due to the long duration required and the lack of effective topical therapies. The lack of a vaccine and better diagnostic tools makes the disease even more difficult to manage and highlights the need for safer and affordable treatments.

Marine sesterterpenoids have diverse structures that are biosynthesized by terpene cyclases and cytochrome P450s from the substrate geranyl farnesyl diphosphate (C_25_) [5,6,7]. Structures comprise a significant diversity of skeletons that are organized as linear, mono-, bi-, tri-, and tetra-carbocyclic sesterterpenes. They display a broad range of bioactivity including cytotoxic, antibacterial, and antiviral activities, selective growth inhibition of hypoxia-adapted cancer cells, and anti-inflammatory activities [8,9,10,11,12,13,14]. There are sesterterpenoids that can be considered as drug leads in the search for new structures to fight pathologies for which effective practice is yet unknown. Although this group is relatively rarer than other terpenoids, many sesterterpenoids have been isolated from marine invertebrates, particularly from marine sponges and their predator nudibranchs [8,9,10,11,12,13,14,15,16,17,18].

This study focuses on marine sponges, a key component of the diverse flora and fauna of the Ryukyu archipelago in southwestern Japan. The Okinawa island group boasts a subtropical climate and unparalleled biodiversity, making it an exceptional site for discovering novel bioactive compounds. In our screening of drug candidates against leishmaniasis in the FT2023-2024 Okinawa Innovation Ecosystem Joint Research Promotion Project, ethyl acetate (EtOAc) extracts of the sponges *Petrosaspongia* sp., *Ircinia* sp., *Luffariella* sp.*,* and the nudibranch *Chromodoris willani* demonstrated >80% growth inhibition of the promastigote *Leishmania major* at 10 μg/mL. By means of bioassay-guided fractionations using open-column chromatography or HPLC, sesterterpenoids 1–3, 4, 5–7, and 8, 9, were purified from *Petrosaspongia* sp., *Ircinia* sp., *Luffariella* sp., and *C. willani*, respectively. By comparing the NMR spectra and the HR-ESI-MS data with those in the literature (Figure 1), three of the four sesterterpenoids (2–4) were identified as hippospongin (2) [17], hipposulfate A (3) [18], and a hippospongin analogue with conjugated γ-tetronic acid 4 [19]. Compound 1, named petrosaspongin, was assumed to be a new analogue of 2 based on NMR spectra. The three sesterterpenoids from *Luffariella* sp. were identified as manoalide (5) [13], secomanoalide (6) [20], and manoalide-25-acetate (7) [21]. The analogues of 5 and 6 from *C. willani* were deoxymanoalide (8) and deoxysecomanoalide (9), thought to be bio-converted by the nudibranch from 5 and 6 found in the prey sponge *Luffariella* sp. [22]. We describe here the structure elucidation of a novel furanosesterterpenoid, petrosaspongin, along with the leishmanicidal activities of eight sesterterpenoids from three marine sponges and a nudibranch collected in Okinawa prefecture, Japan.

## 2. Results and Discussion

### 2.1. Structure Elucidation

Petrosaspongin (**1**) was obtained as a colorless oil with a molecular formula of C_25_H_32_O_4_, established by HR-ESI-MS, which showed an [M + Na]^+^ ion peak at *m/z* 419.2195 (calculated for C_25_H_32_O_4_Na, 419.2193, Δ −0.48 ppm), possessing ten degrees of unsaturation. The NMR data suggest the presence of a β-substituted furan at *δ*_H_ 7.33 (brs, H-1), *δ*_H_ 6.27 (brs, H-2), *δ*_H_ 7.20 (brs, H-4), along with a terminal double bond at *δ*_H_ 4.91 (s), *δ*_H_ 4.93 (s), *δ*_C_ 113.2 (CH_2_-19), *δ*_C_ 144.0 (C-18). A tetronic acid moiety was deduced by the signals at *δ*_H_ 1.71 (s, CH_3_-24), *δ*_C_ 77.3, 176.6, 97.5, 5.9, and 174.8 for C-21 to C-25, also supported by IR spectrum (1749 and 1653 cm^−1^) [17,19,23,24]. The presence of a conjugated triene was proposed at *δ*_H_ 5.48 (t, *J* = 7.0 Hz, H-7), *δ*_H_ 6.15 (d, *J* = 15.3 Hz, H-10), *δ*_H_ 6.33 (dd, *J* = 15.3, 10.8 Hz, H-11), *δ*_H_ 5.86 (d, *J* = 10.8 Hz, H-12) with two vinylic methyl groups at *δ*_H_ 1.76, *δ*_C_ 12.5, (CH_3_-9), *δ*_H_ 1.77, *δ*_C_ 16.7, (CH_3_-14). The HMBC correlations from CH_3_-9 to C-7/C-8/C-10 and CH_3_-14 to C-12/C-13/C-15 also supported the presence of two vinylic methyl groups of CH_3_-9 and CH_3_-14. These were revealed by the HMBC correlations from H-10 to C-7/C-8/C-9/C-11/C-12, from H-11 to C-8/C-10/C-12/C-13, and from CH_3_-9 to C-7/C-8/C-10, consistent with the UV absorption of **1** at 278 nm. The NMR spectra were almost superimposable with those of **2** (Table 1). Hippospongin (**2**) has a disubstituted furan ring fused with a cyclohexene ring, bearing a methyl group and a long chain. The ^1^H and ^13^C NMR spectra of **1** lack the cyclohexene signals of **2** at *δ*_H_ 1.70 (H-6/H-7), *δ*_C_ 38.4 (C-7), *δ*_C_ 38.6 (C-8), and *δ*_C_ 25.8 (C-9), and compound **1** has additional signals of a β-substituted furan ring at *δ*_H_ 7.33 (H-4), a trisubstituted olefin at *δ*_H_ 5.48 (H-7), *δ*_C_ 131.2 (C-7), *δ*_C_ 134.8 (C-8) of the triene, and a vinyl methyl group at *δ*_H_ 1.76 (s, H-9), instead. Detailed analysis of the COSY data revealed five spin systems, as shown in Figure 2a. The spin system of H-1 to H-2 indicated the presence of a β-substituted furan at C-1 to C-4. The HMBC correlations from H-1 and H-2 to C-3 and C-4 supported the presence of a β-substituted furan. Moreover, the lack of cyclohexene signals of **2** fused at C-3 and C-4 of the furane ring suggested that **1** may be a ring-opened isomer of **2**. Without these signals, **1** would harbor the trisubstituted olefin at *δ*_H_ 5.48 (H-7), *δ*_C_ 131.2 (C-7), *δ*_C_ 134.8 (C-8) with the vinyl methyl group at *δ*_H_ 1.76 (s), *δ*_C_ 12.5 (CH_3_-9). This was later supported by the spin system of H-5 to H-7 concomitant with the HMBC correlations from H-6 (*δ*_H_ 2.40) to C-7/C-8/C-3 and from H-5 (*δ*_H_ 2.50) to the furan signals C-2/C-3/C-4 (*δ*_C_ 111.0/124.7/138.9), as shown in Figure 2a. The connectivities between the conjugated triene and the remaining spin systems (H-15 to H-17, H-20 to H-21) were established by HMBC correlations from the vinyl methyl group CH_3_-14 to C-12/C-13/C-15, from the terminal vinyl protons CH_2_-19 to C-17/C-20, and from H-20 (*δ*_H_ 2.27, 2.66) to C-18/C-19/C-25 (*δ*_C_ 174.8) of the terminal tetronic acid (C-21–C-25), consistent with those of **2** (Table 1) [17].

The *E* geometry of Δ^10,11^ double bond was assigned based on the coupling constants (^3^*J*_10,11_ = 15.3 Hz). In ^13^C NMR spectroscopy, the chemical shifts of vinyl methyl groups can provide insight into the geometry of the double bond. Generally, when the ^13^C chemical shift of the methyl group is shifted downfield more than 20 ppm, it suggests the double bond is in a *Z* configuration. Conversely, when the chemical shift of the vinyl methyl group high field-shifted more than 20 ppm, it typically indicates that the double bond is assigned as *E* geometry [25]. The Δ^7,8^ and Δ^12,13^ double bonds were also deduced to have *E* geometries from the ^13^C NMR chemical shifts at *δ*_C_ 12.5 (CH_3_-9) and 16.7 (CH_3_-14). This assignment was further supported by the NOE correlations of CH_3_-9/H-11, CH_3_-14/H-11, H-10/H-12 and H-7, H-12/H-15 in Figure 2b. Consequently, the planar structure of **1** was elucidated as shown in Figure 1. The stereochemistry at C-21 of **1** was assessed using the circular dichroism exciton chirality method; however, no significant cotton effects were observed. Chemical derivatizations were also attempted for **1** by opening the tetronic acid moiety followed by modified Mosher methods. Nonetheless, **1** was easily decomposed under light, oxygen, or heat due to the presence of a furan ring and a triene. The stereochemistry at C-21 remains to be assigned.

### 2.2. Leishmanicidal Activity and Biological Activity

Leishmanicidal activities of isolated sesterterpenoids against *Leishmania major* promastigotes are shown in Table 2. Hippospongin (**2**) exhibited growth inhibition with an IC_50_ 2.8 μM. A new hippospongin analogue, petrosaspongin (**1**) lacking a cyclohexene moiety showed 2.8 times less activity than that of **2**, with hipposulfate A (**3**) having the weakest activity (IC_50_ 88 μM). In addition, compound **4**, the hippospongin analogue with conjugated terminal γ-tetronic acid showed 5.7 times less activity than that of **2**. Since both hipposulfate A without a terminal γ-tetronic acid and compound **4** with the conjugated γ-tetronic acid showed lower activities than **1** and **2**, the key functional group may be a γ-tetronic acid.

Manoalide-25-acetate (**7**) remarkably inhibited the proliferation of *L. major* with an IC_50_ 0.69 μM. Manoalide (**5**) and secomanoalide (**6**) with the presence of a terminal γ-butenolide displayed moderate activities (IC_50_: 11 and 24 μM), while 25-deoxymanoalide (**8**) showed 77 times less activity than that of **7**. These structure–activity relationships suggest that containing γ-butenolide or tetronic moieties may be crucial for sesterterpenoids to exhibit leishmanicidal activity.

Marine sponges are well-known sources for drug discovery, and several publications have suggested their antileishmanial potential [26,27,28]. The majority of anti-leishmanial compounds from marine sponges are related to alkaloids [28]. Alkaloid pseudoceratidines demonstrated moderate activity against *Leishmania amazonesis* and *L. infantum* promastigotes with IC_50_ 19–76 μM [29]. Other alkaloids, monalidines and batzelladines, have relatively potent activity at IC_50_ 2–4 μM against *L. infantum* promastigotes [30]. A few papers were recently published on leishmanicidal active sesterterpenoids. Furanosesterterpenoid, ircinin-1, showed leishmanicidal activities against *L. donovani* (IC_50_ range: 28–130 μM) [31], and Majer et al. reported ircinianin-type sesterterpenoids exhibiting moderate leishmanicidal activity against *L. donovani* (IC_50_ 16.6 μM) [32]. Although we cannot simply compare those published IC_50_ values with our results, it is noteworthy that the IC_50_ value 0.69 μM of manoalide-25-acetate is the most potent leishmanicidal activity among the marine-derived sesterterpenoids.

The structure–activity relationships of isolated sesterterpenoids suggest that the γ-lactone would be a key functionality. Although we have no experimental evidence for the molecular target for these sesterterpenoids, the clerodane diterpenoid possessing a γ-butenolide ring, 16α-hydroxycleroda-3,13 (14) *Z*-diene15,16-olide (clerodane) was reported as an inhibitor of DNA topoisomerase type I encoded in *L. donovani* (Ld-topoI). The clerodane inhibited catalytic activity of the recombinant Ld-topoI, which is essential for parasite growth, and ultimately, induced apoptosis. Molecular docking experiments with the clerodane showed five strong hydrogen-bonding interactions and hydrophobic interactions with Ld-topoI. Notably, the γ-butenolide ring of clerodane forms strong hydrogen bonds with the NH group of amino acid residue Arg314, oxygen of Asn452 and oxygen of Cys452 at the active site of Ld-topoI [33]. To gain more insight into the interactions between the sesterterpenoid γ-lactones with the active site of DNA topoisomerase I in *Leishmania* sp., we conducted a molecular docking simulation using the free docking Web service SwissDock [34]. Since the γ-butenolide ring of clerodane was reported as a crucial functional group for the inhibition of Ld-topI, we used the X-ray crystal data of Ld-topoI (PDB code: 2B9S) as the target molecule [35]. Seven compounds containing γ-lactone were used as a ligand. AutoDock Vina 1.2.0, facilitating the design and execution of a simple docking simulation, was used for the calculation algorithm to generate the docking results of ligand and Ld-topI (Appendix A) [34]. As a result, the oxygens of the γ-lactone ring have at least one or two hydrogen bonding interactions with several amino acid residues at active sites of Ld-topI. Although this molecular docking simulation cannot conclude the specific molecular target, the hydrogen bonding interactions at the γ-lactone group may contribute to showing the leishmanicidal activities of those sesterterpenoids. The mechanism of action of sesterterpenoids possessing γ-lactones should be assessed by further investigations on the interaction of γ-lactone sesterterpenoids with DNA topoisomerase of *L. major*.

The nonprofit organization, Drugs for Neglected Diseases Initiative (DNDi), suggests that a selectivity index with cytotoxicity against a human hepatoma cell line HepG2 should be at least >5 as a first criterion for in vitro screening of drug candidates for cutaneous leishmaniasis [36]. To know the selectivity of manoalide-25-acetate (**7**), the cytotoxicity of **7** against HepG2 was demonstrated and did not show any cytotoxicity (the highest concentration was >20 μM for 48 h treatment, not tested with higher concentrations). The selectivity index of **7** is 28 times higher than that recommended by the DNDi. However, manoalide-25-acetate (**7**) exhibits potent cytotoxicity against cancer cell lines with IC_50_ = 0.26, 0.63, 0.76, 1.68 μM for MCF-7 (a human breast adenocarcinoma), KB (human oral epidermoid carcinoma), HT-29 (colorectal carcinoma), and HeLa (human cervical carcinoma), respectively [37]. It would therefore be inappropriate as a drug for visceral leishmaniasis with oral or intra-abdominal administration, though it would be possible to develop it as a topical treatment for cutaneous leishmaniasis. Manoalide-25-monoacetate exhibited the most potent activity against *L. major* promastigotes but not as high potency as other sesterterpenoids with hydroxy or dehydrated γ-butenolides or tetronic acids. This result suggested that acetylation on γ-butenolide could become a driving force for binding its target molecules or the permeability of lipophilic compounds. Manoalides are well-studied for structure–activity relationships against their molecular target, phospholipase A_2_ (PLA2) in mammalian cells. Faulkner et al. reported that γ-hydroxy butenolide ring is involved in the initial interaction of manoalide with PLA2 and irreversibly inhibits PLA2 [13,14]. Further studies on the mechanism of toxicity in *Leishmania* genus are ongoing.

## 3. Materials and Methods

### 3.1. General Experimental Procedures

^1^H, ^13^C and 2D NMR spectra were recorded on Bruker Avance III 500 spectrometer (500 MHz for ^1^H NMR,125 MHz for ^13^C NMR). Chemical shifts are denoted in *δ* (ppm) relative to the residual solvent peaks as internal standards (CDCl_3_, δ_H_ 7.26, δ_C_ 77.0). Data for NMR spectra were reported as follows: chemical shifts (δ) in ppm; s, singlet; d, doublet; t, triplet, m, multiplet; br, broad signal; *J*, coupling constants in Hz. Data were analyzed using Topspin 4.1.4. ESI-MS spectra were recorded on a JEOL JMS-T1000GCV spectrometer. Optical rotations were recorded on a JASCO P-1010 polarimeter. All reagents were used as supplied, unless otherwise stated. Column chromatography was performed using silica gel 60 (Merck); thin layer chromatography (TLC), silica gel 60 F254 plates (Merck). High-performance liquid chromatography (HPLC), Hitachi L-6000 pump fitted with a Hitachi L-4000 UV monitor and a Shodex RI-101 monitor to detect compounds. A 5C_18_-AR-II column (Cosmosil) or a 5SL-II column (Cosmosil) was used for reversed-phase or normal-phase HPLC, respectively. The details of the conditions for HPLC are described below.

### 3.2. Preparation of Extracts from Marine Organisms

An extract library of marine organisms established in another project from 2009 to 2014 and in 2023 was used for the screening in the FT2023-2024 Okinawa Innovation Ecosystem Joint Research Promotion Project. The organisms included sponges, soft corals, and algae, hand-collected during SCUBA or rebreather diving from coral reefs in the Ryukyu archipelago near Okinawa, Kerama, Kume, Miyako, Iriomote, and Yonaguni islands of Okinawa prefecture. The specimens were brought back to the lab after collection in fresh or frozen condition and were kept frozen until extraction. The general extraction procedure involved steeping a thawed specimen in acetone three times and the combined acetone solutions were then filtered and concentrated in vacuo. The residual material was partitioned between EtOAc and water. The organic layer was concentrated to give an EtOAc extract, while the aqueous layer was concentrated to a dry material which was washed with a small amount of methanol (MeOH). The MeOH solution was then concentrated to give a MeOH extract. Both EtOAc and MeOH extracts were kept at −30 °C for screening. The sponges were identified by one of us (N.J.dV.).

### 3.3. Isolation of Sesterterpenoids

The extracts that exhibited >80% growth inhibition at 10 μg/mL against *Leishmania major* promastigotes were examined further. One EtOAc bioactive extract was prepared from the sponge *Petrosaspongia* sp. (464 g, wet, registration specimen code at Naturalis: RMNH. POR.12476), collected off Kume Island, Okinawa prefecture in September 2009 (26°20′05′′ N 126°44′08′′ E, depth: 45 m). A portion of the EtOAc extract (241.2 mg of 7.2 g) was subjected directly to reversed-phase HPLC (Cosmosil 5C_18_-AR-II column; *ϕ* 10 × 250 mm) eluted in isocratic mode with solvent, acetonitrile (CH_3_CN), in water at 90% with 0.1% formic acid (HCOOH), to obtain as major compounds petrosaspongin (**1**) [46.2 mg; retention time (Rt) = 10.0 min], hippospongin (**2**) (48.0 mg; Rt = 10.7 min), and hipposulfate A (**3**) (22.1 mg; Rt = 7.5 min).

Another EtOAc extract was prepared from the marine sponge *Ircina* sp. (206 g, wet, specimen code; RMNH. POR.12475) collected by hand off Iriomote island, Okinawa prefecture, Japan in July 2012 (24°20′20” N 123°41′37” E, depth: 60 m). A portion of the EtOAc extract (52.0 mg of 3.8 g) was purified by reversed-phase HPLC (Cosmosil 5C_18_-AR-II column; *ϕ* 10 × 250 mm) eluted in isocratic mode with solvent CH_3_CN in water at 90% with 0.1% HCOOH, to obtain compound **4** (4.2 mg; Rt = 11.8 min).

An EtOAc extract (642 mg) of the fresh sponge *Luffariella* sp. (51.5 g, wet, specimen code; RMNH. POR.12477) collected at Manza, Okinawa island in July 2023 (26°30′17”N 127°50′38”E, depth: 28 m), was subjected to open column chromatography on silica gel eluted with a stepwise gradient of solvents (*n*-hexane, 90%, 75% and 33% of dichloromethane in EtOAc, and 100% EtOAc), resulting in seven fractions (1–7). Fraction 3 (4.3 mg of 226.5 mg) was further purified by normal-phase HPLC on a Cosmosil 5SL-II column (*ϕ* 4.6 × 250 mm), eluted with 78% *n*-hexane in EtOAc, to yield manoalide (**5**) (2.8 mg; Rt = 13.0 min). Fraction 5 (4.0 mg from 75.6 mg) was purified on normal-phase HPLC (Cosmosil 5SL-II column; *ϕ* 4.6 × 250 mm) in isocratic mode with 75% *n*-hexane in EtOAc, yielding secomanoalide (**6**) (1.2 mg; Rt = 12.0 min). Manoalide-25-acetate (**7**) (22.1 mg; Rt = 5.0 min) was purified from fraction 1 (120.0 mg) with normal-phase HPLC on a 5SL-II column (*ϕ* 4.6 × 250 mm, eluted with 80% *n*-hexane in EtOAc).

The EtOAc extract (4.0 mg) of the nudibranch *C. willani* (1.1 g, wet, specimen code; RU114N) collected with the aforementioned prey sponge *Luffariella* sp., was directly subjected to HPLC on a Cosmosil 5SL-II column (*ϕ* 4.6 × 250 mm), eluted with 75% *n*-hexane in EtOAc, to yield deoxymanoalide (**8**) (0.9 mg; Rt = 8.0 min). The fraction washed with 100% EtOAc was further separated with the same column eluted with 60% of *n*-hexane in EtOAc to yield deoxysecomanoalide (**9**) (0.1 mg; Rt = 11.0 min).

Petrosaspongin (**1**): Colorless oil; ${\left[\alpha \right]}_{D}^{25}$ +18.3° (*c* 0.08, CHCl_3_); HR-ESI-MS: *m/z* 419.2195 [M+Na]^+^ (calcd. for C_25_H_32_O_4_Na, 419.21928, Δ -0.48 ppm), IR (film): *ν*_max_ 1749, 1653 cm^−1^; UV (CH_3_CN) *λ*_max_ (log *ε*): 278 (3.9), 228 nm (4.1). The ^1^H and ^13^C NMR (CDCl_3_) spectra are shown in Table 1 and Appendix A.

### 3.4. In Vitro Leishmanicidal Assay

The leishmania growth medium for cutaneous-type promastigotes of *Leishmania major* (MHOM/SU/73/5ASKH) consisted of Medium-199 supplemented with 10% fetal calf serum, 100 IU/mL penicillin, and 100 µg/mL streptomycin. Promastigotes were cultured at 27 °C and 5% CO_2_ in an incubator.

The leishmanicidal effects of the extracts and isolated compounds were assessed using the improved 3-[4,5-dimethylthiazol-2-yl]-2,5-diphenyl- tetrasodium bromide (MTT) method as follows: cultured promastigotes were seeded at 5 × 10^4^/50 μL of medium per well in 96-well microplates. Then 50 μL of varying concentrations of test compounds dissolved in a mixture of dimethyl sulfoxide (DMSO) and the medium were added to each well. Each concentration was tested in triplicate. The microplate was then incubated at 27 °C in 5% CO_2_ for 48 h, following which 10 µL of the Cell Counting Kit-8 (Dojindo, Japan) was added to each well and the plates further incubated at 27 °C for 6 h. Optical density values were measured using an ARVO MX microplate reader (PerkinElmer Japan Co., Ltd., Kanagawa, Japan), with a test wavelength of 450 nm and a reference wavelength of 595 nm.

The IC_50_ values were calculated by fitting the data to a non-linear regression using a dose-response inhibitory model in Microsoft Excel, with experiments conducted in triplicate (n = 3). Amphotericin B served as a positive control (IC_50_ 0.10 µM).

### 3.5. Molecular Docking Simulation

The target X-ray crystal data of the type I DNA topoisomerase of *Leishmania donovani* (Ld-topI) was downloaded from the Protein Data Bank (PDB code: 2B9S) [35]. The nucleic acids were removed from 2B9S data and saved with PyMOL software (PyMOL Molecular Graphics System V4.6.0, Schrödinger, LLC., New York, USA) for further experiments. The 2D chemical structures of selected sesterterpenoids, petrosaspongin (**1**), hippospongin (**2**), compound **4**, manoalide (**5**), secomanoalide (**6**), and manoalide-25-acetate (**7**) were sketched with ChemDraw 20.1.1 and submitted as a ligand in SwissDock to generate a SMILES file. The calculation program, “Docking with AutoDock Vina” was chosen; https://www.swissdock.ch/ [34]. To submit the crystal data to the web server, the double strand DNA was deleted from original 2B9S crystal data and uploaded as a target molecule. To run docking simulation in SwissDock, we had to define the calculation area. The search box center and size were defined as [Box center: 33-50-8 Å, Box size: 25-25-25 Å], which can cover the active site pocket of DNA topoisomerase type I. The number of samplings was set to “10” and generated a docking result for seven ligands. All results are shown in Appendix A.

### 3.6. Cytotoxicity of Manoalide-25-Acetate

HepG2 cells were seeded in 96-well plates at a density of 2.5 × 10^4^ cells/well. The following day, the cells were cultured with manoalide-25-acetate (**7**) for 48 h, and cytotoxicity was determined using the WST-8 assay with the Cell Counting Kit-8 (Dojindo).

## 4. Conclusions

Nine sesterterpenoids were isolated through bioassay-guided fractionations from three different Okinawan marine sponges and a nudibranch. The planar structure of the previously undescribed sesterterpenoid petrosaspongin (**1**) was determined by HR-ESI-MS and detailed NMR analysis. The geometries at double bonds were elucidated by chemical shifts of vinyl methyl groups, coupling constants, and NOESY correlations. Isolated sesterterpenoids that harbored γ-tetronic acid or γ-butenolide groups exhibited leishmanicidal activities, with IC_50_ values from 0.69 to 53 μM. The sesterterpenoids exhibited noteworthy bioactivity, with manoalide-25-acetate (**7**) showing the highest potency. Further research will be required to elucidate the mode of actions of sesterterpenoids with γ-lactones. The findings underscore the importance of exploring marine biodiversity for novel therapeutic agents, suggesting that further investigations into the mechanisms of action and structural optimization of these compounds could lead to effective and safer treatments for leishmaniasis. This research adds to the understanding of the bioactive potential of marine organisms, and also addresses a critical need in public health for accessible and effective therapies against neglected tropical diseases.

## Figures and Tables

**Figure 1 marinedrugs-23-00016-f001:**
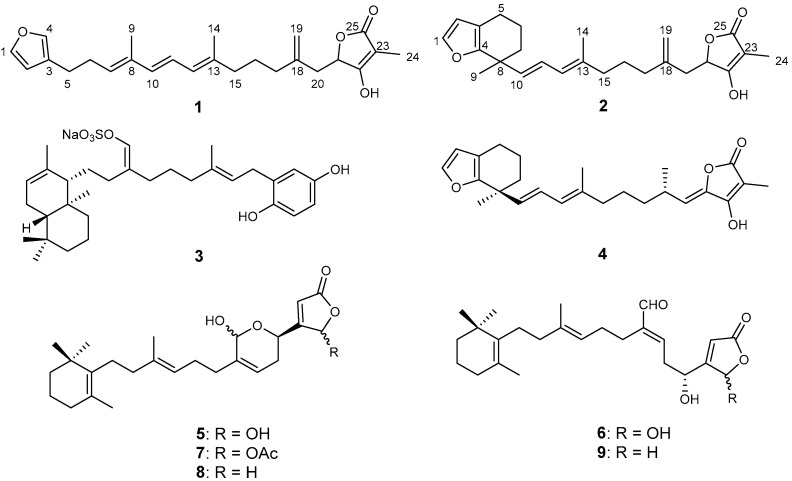
The structures of compounds (**1**–**9**).

**Figure 2 marinedrugs-23-00016-f002:**
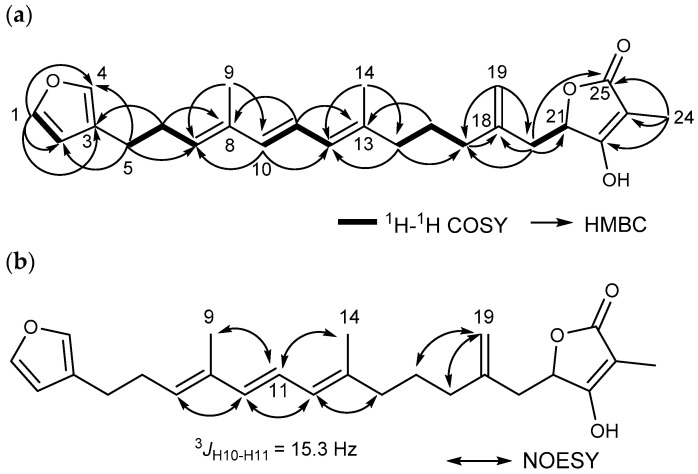
Key 2D NMR correlations for petrosaspongin (**1**). (**a**) ^1^H-^1^H COSY and key HMBC correlations. (**b**) Key NOESY correlations.

**Table 1 marinedrugs-23-00016-t001:** NMR data of petrosaspongin (**1**) and hippospongin (**2**).

Position	1 ^a^		2 [17]	
	*δ* _C,_ Type	*δ* _H_ (*J* in Hz)	*δ* _C,_ Type	*δ* _H_ (*J* in Hz)
1	142.7, CH	7.33, brs	140.6, CH	7.24, d (1.8)
2	111.0, CH	6.27, brs	110.1, CH	6.16, d (1.8)
3	124.7, C	-	116.7, C	-
4	138.9, CH	7.20, brs	154.4, C	-
5	24.8, CH_2_	2.50, m	22.5, CH_2_	2.40, m
6	28.9, CH_2_	2.40, m	20.0, CH_2_	1.70, m
7	131.2, CH	5.48, t (7.0)	38.4, CH_2_	1.70, m
8	134.8, C	-	38.6, C	-
9	12.5, CH_3_	1.76, s	25.8, CH_3_	1.34, s
10	135.5, CH	6.15, d (15.3)	138.8, CH	5.60, d (15.2)
11	123.0, CH	6.33, dd (15.3, 10.9)	124.9, CH	5.95, dd (15.2, 10.6)
12	125.6, CH	5.86, d (10.9)	124.8, CH	5.77, d (10.6)
13	137.8, C	-	137.0, C	-
14	16.7, CH_3_	1.77, s	16.5, CH_3_	1.63, s
15	39.5, CH_2_	2.06, m	39.3, CH_2_	2.00, m
16	25.8, CH_2_	1.58, m	25.8, CH_2_	1.50, m
17	35.9, CH_2_	2.06, m	35.9, CH_2_	2.00, m
18	144.0, C	-	143.8, C	-
19	113.2, CH_2_	4.91, s	113.0, CH_2_	4.89, s
		4.93, s		4.91, s
20	38.3, CH_2_	2.27, dd (15.1, 8.5)	38.2, CH_2_	2.26, dd (15.0, 8.2)
		2.66, dd (15.1, 3.7)		2.62, dd (15.0, 3.9)
21	77.3, CH	4.83, dd (8.5, 3.7)	77.8, CH	4.79, dd (8.2, 3.9)
22	176.6, C	-	177.5, C	-
23	97.5, C	-	97.0, C	-
24	5.9, CH_3_	1.71, s	5.9, CH_3_	1.69, s
25	174.8, C	-	175.6, C	-

^a 1^H (500 MHz) and ^13^C (125 MHz) NMR data in CDCl_3_.

**Table 2 marinedrugs-23-00016-t002:** IC_50_ of isolated compounds against promastigotes *L. major*.

Compounds	IC_50_ (μM)
petrosaspongin (**1**)	7.8
hippospongin (**2**)	2.8
hipposulfate A (**3**)	88
compound **4**	16
manoalide (**5**)	24
secomanoalide (**6**)	11
manoalide-25-acetate (**7**)	0.69
deoxymanoalide (**8**)	53
deoxysecomanoalide (**9**)	3.0
amphotericin B	0.10

## Data Availability

Data are contained within this article and Appendix A.

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
