# Peer review of "A Novel Sesterterpenoid, Petrosaspongin and γ-Lactone Sesterterpenoids with Leishmanicidal Activity from Okinawan Marine Invertebrates"

_marinedrugs, 2024, doi:10.3390/md23010016_

Round 1
Reviewer 1 Report
Comments and Suggestions for Authors
Finding new types of leishmanicidal compounds is a critical endeavor for the treatment of Leishmaniasis. The authors conducted the chemical and biological investigations of three sponges and one nudibranch from Okinawa. As a result, an unreported furanosesterterpenoid and eight known sesterterpenoids were obtained. The structure of new compound were elucidated by the analysis of NMR data and comparison with those of a known analogue. Interestingly, all the isolateds exhibited moderate different levels of leishmanicidal activity. Based on these important findings, this work is suggested to be published in the forthcoming issue of this journal.
However, revisions were required as followings:
1. P3L108: The datum ‘Δ -0.48 ppm’ was not consistent with ‘Δ -0.52 ppm’(P8L299).
2. P3L110: The data δH 4.91 (s), δH 4.93 (s), δC 113.2 (CH2-19), δC 144.0 (C-18) were referred to a terminal double bond, not terminal vinyl protons.
3. P3L117: The HMBC correlations from CH3-129 to C-12/C-13/C-15 should be added to confirm the presence of a vinylic methyl group CH3-14.
4. For compound 2, the furan ring fused with a cyclohexene ring, bearing a methyl and a long chain. The description in the manuscript was inappropriate.
5. P3L124: The spin system was H-1 to H-2 not H-1 to H-4. Moreover, the HMBC correlations from H-1 and H-2 to C-3 and C-4 should be describe to support the presence of a β-substituted furan at C-1 to C-4.
6. For the stereochemistry of C-21, did you try the quantum mechanical-nuclear magnetic resonance (QM-NMR) method? It is a powerful tool for the determination of configuration.
7. In this work, it is interesting to find that compounds 8 and 9 from the nudibranch Chromodoris willani were bio-converted products of compounds 5 and 6 found in the prey sponge Luffariella sp. Was the predator–prey relationship for the sponge and nudibranch confirmed by the researchers previously? And what are the benefits of these two compounds for the nudibranch?
8. The specimen numbers of three sponges and one nudibranch should be provided.
9. The retention times (Rt) should be provided for the compounds finally purified by HPLC in the subsection ‘3.3. Isolation of Sesterterpenoids’.
10. P4L155: ‘H12 / H-15’ → ‘H-12 / H-15’
Author Response
All changes in the main text are highlighted in red.
- P3L108:The datum ‘Δ -0.48 ppm’ was not consistent with ‘Δ -0.52 ppm’(P8L299).
Response: Corrected (Δ -0.48 ppm) in P3L101 and P8L316. - P3L110: The dataδH 4.91 (s), δH 4.93 (s), δC 113.2 (CH2-19), δC 144.0 (C-18) were referred to a terminal double bond, not terminal vinyl protons.
Response: Corrected as indicated in P3L103. - P3L117: The HMBC correlations from CH3-129 to C-12/C-13/C-15 should be added to confirm the presence of a vinylic methyl group CH3-14.
Response: Inserted the sentence in P3L109 ” The HMBC correlations from CH3-9 to C-7/C-8/C-10 and CH3-14 to C-12/C-13/C-15 also supported the presence of two vinylic methyl groups of CH3-9 and CH3-14. ” to support the presence of the vinylic methyl groups. - For compound2, the furan ring fused with a cyclohexene ring, bearing a methyl and a long chain. The description in the manuscript was inappropriate.
Response: Corrected by the insertion of the sentence in P3L113 “Hippospongin (2) has a disubstituted furan ring fused with a cyclohexene ring, bearing a methyl group and a long chain. The 1H and 13C NMR spectra of 1 lack the cyclohexene signals of 2 at δH 1.70 (H-6/H-7), δC 38.4 (C-7), δC 38.6 (C-8), and δC 25.8 (C-9), and com….”. - P3L124: The spin system was H-1 to H-2 not H-1 to H-4. Moreover, the HMBC correlations from H-1 and H-2 to C-3 and C-4 should be describe to support the presence of a β-substituted furan at C-1 to C-4.
Response: Corrected the spin system from H-1/H-4 to H-1 to H-2. Added the sentence “The HMBC correlations from H-1 and H-2 to C-3 and C-4 supported the presence of a β-substituted furan.” in P3L121. - For the stereochemistry of C-21, did you try the quantum mechanical-nuclear magnetic resonance (QM-NMR) method? It is a powerful tool for the determination of configuration.
Response: We appreciate your suggestion. We are not familiar with the QM-NMR method for determining stereochemistry. We did not apply compound 1 to the QM-NMR method. Our institute is currently not available for QM-NMR. Besides, due to compound 1 being quite unstable, we have to re-isolate from the extract just before the measurement. Even though we could find a collaborator who is familiar with the method, it would be difficult to measure QM-NMR before the decomposition of 1. - In this work, it is interesting to find that compounds 8 and 9 from the nudibranch Chromodoris willani were bio-converted products of compounds 5 and 6 found in the prey sponge Luffariella sp. Was the predator–prey relationship for the sponge and nudibranch confirmed by the researchers previously? And what are the benefits of these two compounds for the nudibranch?
Response: The predator-prey relationship for the sponge and nudibranch was confirmed by the researcher [Ref. 22] as well as Prof. Faulkner et al, [Kernan M. R., Barrabee E. B., Faulkner D. J., Comp. Biochem. Physiol., 89B, 275—278 (1988)]. According to Ref. 22, this bio-conversion has not been proven experimentally, although a similar conversion has been reported between the deoxy analogues from the nudibranch Chromodoris funerae and the manoalide analogues, luffarriellins, from the sponge L. variabilis [Kernan M. R., Comp. Biochem. Physiol., 89B, 275 (1988)]. Manoalides are known to be repellent substances to avoid predation from other marine organisms such as fishes or clubs. Although the benefits of these deoxy-manoalides are still unknown, this bioconversion may contribute to how nudibranchs avoid self-toxicity caused by accumulated cytotoxins. Since γ-butenolide of manoalides irreversibly inhibits phospholipase A2, it could show potent cytotoxicity not only against fish but also against nudibranchs. These deoxy-manoalides may play a role in the resistance mechanism of nudibranchs. - The specimen numbers of three sponges and one nudibranch should be provided.
Response: Added the specimen codes in the subsection 3.3. Isolation of Sesterterpenoids. - The retention times (Rt) should be provided for the compounds finally purified by HPLC in the subsection ‘3.3. Isolation of Sesterterpenoids’.
Response: Added the retention times of all isolated compounds as indicated. - P4L155:‘H12 / H-15’ → ‘H-12 / H-15’
Response: Corrected.
Reviewer 2 Report
Comments and Suggestions for Authors
This article describes nine sesterterpenoids isolated from okinawan marine invertebrates with leishmanicidal activity, these findings contribute to the ongoing search for more effective treatments against CL, please the author provide brief answers to the following questions.
Q1:Is the summary completely finished? No closing punctuation in the last sentence of the abstract.
Q2: Excessive space is devoted to cutaneous leishmaniasis (CL) in the introduction, but the mechanism of action for compound 1 is not elaborated, please rationalize or streamline the introduction section?
Q3: For compound 1, the article mentions that the stereochemistry at C-21 cannot be determined by circular dichroism exciton chirality method and chemical derivatizations. Could the author attempt other approaches, such as optical rotatory dispersion (ORD)?
Q4: In General experimental procedures, there is no mention of the type of column and this should be added.
Q5: Retention times of the isolated compounds have not been raised in Isolation of Sesterterpenoids and the author should complete this section.
Comments on the Quality of English Language
The English could be improved to more clearly express the research.
Author Response
- Q1:Is the summary completely finished? No closing punctuation in the last sentence of the abstract.
Response: Corrected. - Q2: Excessive space is devoted to cutaneous leishmaniasis (CL) in the introduction, but the mechanism of action for compound 1 is not elaborated, please rationalize or streamline the introduction section?
Response: The sentence about the mechanism of action has been removed and remain those sentences for the problems of current drugs. - Q3: For compound 1, the article mentions that the stereochemistry at C-21 cannot be determined by circular dichroism exciton chirality method and chemical derivatizations. Could the author attempt other approaches, such as optical rotatory dispersion (ORD)?
Response: We have already tried to get ECD or ORD data four times, unfortunately, the results were not clear enough to proceed with the calculation study. - Q4: In General experimental procedures, there is no mention of the type of column and this should be added.
Response: Added the type of column and names in general experimental procedures in L259. - Q5: Retention times of the isolated compounds have not been raised in Isolation of Sesterterpenoids and the author should complete this section.
Response: Added the retention times of all isolated compounds as indicated. - The English could be improved to more clearly express the research.
Response: The English proofreading was done by a native speaker twice, who has professional experience in English proofreading since the 1990s.
Round 2
Reviewer 1 Report
Comments and Suggestions for Authors
All the issues have been addressed by the authors, and this manuscript could be accepted now.
Reviewer 2 Report
Comments and Suggestions for Authors
The problems have been modified and is recommended to be accepted.
Comments on the Quality of English Language
The problems have been modified and is recommended to be accepted.